# Organic Thin-Film Transistors as Gas Sensors: A Review

**DOI:** 10.3390/ma14010003

**Published:** 2020-12-22

**Authors:** Marco Roberto Cavallari, Loren Mora Pastrana, Carlos Daniel Flecha Sosa, Alejandra Maria Rodriguez Marquina, José Enrique Eirez Izquierdo, Fernando Josepetti Fonseca, Cleber Alexandre de Amorim, Leonardo Giordano Paterno, Ioannis Kymissis

**Affiliations:** 1Engenharia de Energia, Universidade Federal da Integração Latino-Americana, Foz do Iguaçu, PR 85866-000, Brazil; cdf.sosa.2017@aluno.unila.edu.br (C.D.F.S.); amr.marquina.2017@aluno.unila.edu.br (A.M.R.M.); 2Departamento de Engenharia de Sistemas Eletrônicos, Escola Politécnica da Universidade de São Paulo, São Paulo, SP 05508-010, Brazil; lmora85@usp.br (L.M.P.); jeeizquierdo@usp.br (J.E.E.I.); fjfonseca@usp.br (F.J.F.); 3Departamento de Engenharia de Biossistemas, Faculdade de Ciências e Engenharia (Câmpus de Tupã), Universidade Estadual Paulista Júlio de MesquitaFilho (UNESP), Tupã, SP 17602-496, Brazil; cleber.amorim@unesp.br; 4Laboratório de Pesquisa em Polímeros e Nanomateriais, Instituto de Química, Universidade de Brasília, Brasília, DF 70910-900, Brazil; lpaterno@unb.br; 5Department of Electrical Engineering, Columbia University, New York, NY 10027-6902, USA; johnkym@ee.columbia.edu

**Keywords:** flexible electronics, organic semiconductors, organic thin-film transistors, organic field- effect transistors, gas sensors

## Abstract

Organic thin-film transistors (OTFTs) are miniaturized devices based upon the electronic responses of organic semiconductors. In comparison to their conventional inorganic counterparts, organic semiconductors are cheaper, can undergo reversible doping processes and may have electronic properties chiefly modulated by molecular engineering approaches. More recently, OTFTs have been designed as gas sensor devices, displaying remarkable performance for the detection of important target analytes, such as ammonia, nitrogen dioxide, hydrogen sulfide and volatile organic compounds (VOCs). The present manuscript provides a comprehensive review on the working principle of OTFTs for gas sensing, with concise descriptions of devices’ architectures and parameter extraction based upon a constant charge carrier mobility model. Then, it moves on with methods of device fabrication and physicochemical descriptions of the main organic semiconductors recently applied to gas sensors (i.e., since 2015 but emphasizing even more recent results). Finally, it describes the achievements of OTFTs in the detection of important gas pollutants alongside an outlook toward the future of this exciting technology.

## 1. Introduction

Organic electronics is a branch of modern electronics dealing with electronic devices based on electroactive organic materials, including carbonaceous nanomaterials, conjugated polymers and small molecules. Indeed, carbon-based materials are the closest to the biomolecules of living things [1]. Conventional organic materials have been used in electronic applications for more than a century for insulation or protection purposes in a number of applications. However, the discovery of electrical conductivity in trans-poly(acetylene) in 1976 [2] opened up a new venue for organic materials in the field of electronics. This new field rapidly evolved in the 1990s after realization of the first organic light emitting diodes (OLEDs) [3], followed by the controlled synthesis of fullerenes [4] and their employment in the first organic photovoltaic (OPVs) cells [5]; the development of carbon nanotubes (CNTs) [6]; and finally, the measurement of electrical properties of single-layer graphene in later 2004 [7]. The most important dates and achievements in organic electronics are summarized in the timeline given in Figure 1. On the one hand, the scientific community has recognized the importance of these special organic materials with two Nobel prizes in a decade—the first, in Chemistry, in 2000, for the development of conjugated polymers, and the second, in Physics, in 2010, for the measurement of the electrical properties of graphene. Organic semiconductors make possible several practical applications—OLEDs, OPVs, chemosensors [8,9,10], supercapacitors [11], transistors [12], radiation detectors [13,14] and so on—while offering, at much lower cost, several advantages over well-established inorganic materials, including low-weight, mechanical flexibility and lower energy consumption.

Organic thin-film transistors (OTFTs), the main subject of this comprehensive review, were developed in the 1980s. The first successful attempt was published by Ebisawa et al. [15], wherein they showed a metal–insulator–semiconductor field-effect-transistor (MISFET) with a poly(acetylene)/poly(siloxane) interface working as a depletion-type transistor. This device, however, exhibited very low transconductance (ca. 13 n$\Omega^{-1}$) and a slow response time. Kudo et al. [16] demonstrated an increase in the photoelectric quantum efficiencies with respect to an increase in dye charge carrier mobility (μ). Tsumura et al. [17] built an OTFT with poly(thiophene) as the semiconductor, allowing the semiconductor conductivity to be modulated by a factor of $10^{2}$ to $10^{3}$. In the same work, they also suggested that macromolecules were promising materials for these new electronic devices. In 1988, Clarisse et al. demonstrated the first small molecule-based TFT [18]. The transistor had a current modulation ($I_{ON/OFF}$) of $10^{3}$ and even higher conductivity when compared to semiconducting polymers. This paper was an early demonstration that superior electrical performance should be achieved by using single crystals from these small organic molecules.

The following decades have witnessed the increase in charge carrier mobility in OTFTs as a result of new processing methods and the synthesis of myriad new small-molecule semiconductors, polymeric semiconductors and carbonaceous nanomaterials. Mobility increased from $10^{-3}$ cm${}^{2}$/V·s in a MISFET processed by successive vacuum evaporation of alpha-conjugated sexithienyl in 1989 [19] to 0.08 cm${}^{2}$/V·s in an n-channel FET using a fullerene (C60) with current modulation of $10^{6}$ in 1995 [20]. In 1997, Lin et al. achieved a μ of 1.5 cm${}^{2}$/V·s in a pentacene-based transistor [21]. This OTFT displayed an $I_{ON/OFF}$ greater than $10^{8}$ and a subthreshold slope of less than 1.6 V. It had the largest field-effect mobility and smallest subthreshold slope published at that time, overcoming amorphous-silicon-based TFT technology. Further breakthroughs were related to the demonstration of TFTs integrating other carbon nanomaterials, such as nanotubes and graphene. In 1998, Tans et al. [22] reported the fabrication of a field-effect transistor consisting of just one semiconducting single-wall CNT. The fabrication of the three-terminal switching device at the level of a single molecule represented an important step towards molecular electronics, although integration into a circuit was still a challenge. Depending upon the nanotube’s chirality (armchair, chiral or zigzag, as shown in Figure 1), it featured conducting or semiconducting electrical characteristics [23,24]. A new breakthrough was achieved with graphene in 2004, when a strong ambipolar electric field effect was observed, reaching mobilities of approximately 10,000 cm${}^{2}/$V·s [7]. Unfortunately, these devices showed almost no current modulation under DC current versus voltage measurements ($I_{ON/OFF}$ < 30). Despite years of continuous improvements, the OTFTs displaying larger current modulations reported more recently [25] reach a charge carrier mobility of ca. 1–10 cm${}^{2}$/V·s, which is about 100 to 1000 times lower than that achieved with single-crystal silicon. Nevertheless, such mobilities are high enough for applications such as backplanes for flexible active-matrix organic light-emitting displays (AMOLEDs) [26] and flexible electronic paper [27]. Recent demonstrations of circuit applications include the integration of an organic line driver for an AMOLED display [28] and an organic microprocessor [25].

**Figure 1 materials-14-00003-f001:**
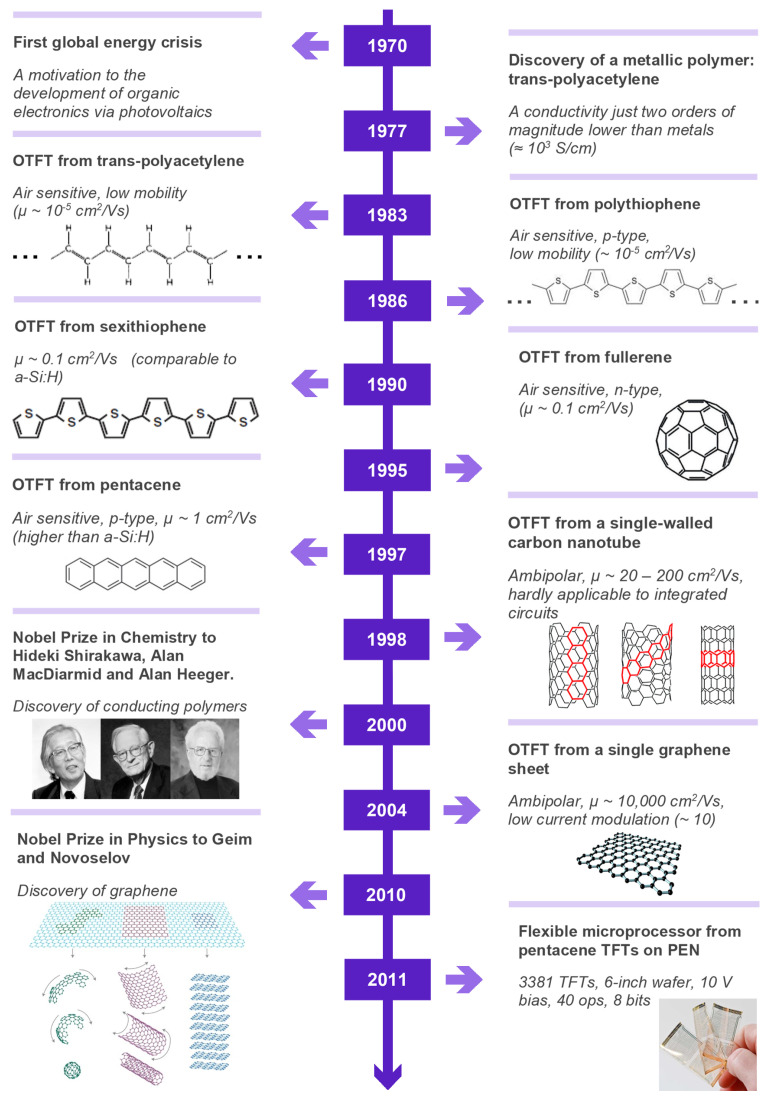
Timeline featuring the main events related to the development of organic electronics. The chemical structures of carbon nanotubes: reprinted from [23]; published by The Royal Society of Chemistry. The chemical structure of graphene: reprinted from reference [29] with permission from Elsevier. Photographs of A. J. Heeger, A. G. McDiarmid and H. Shirakawa 2000 Nobel Prize in Chemistry recipients: reproduced from reference [30] with permission from The Royal Society of Chemistry. Graphene as a 2D building material for carbon materials of all other dimensionalities: reprinted by permission from Springer Nature Customer Service Centre GmbH: Nature, Nature Materials [31], ©2007. The photograph of flexible microprocessors: ©2012 IEEE. Reprinted, with permission, from [25].

In 2011, Myny et al. used OTFT technology to develop an 8-bit pentacene microprocessor on plastic foil, with a limit of 40 operations per second and a power consumption as low as 100 μW operating at a supply voltage of 10 V and a back-gate voltage of 50 V [25]. This microprocessor could execute user-defined programs such as a calculator, a timer or even a game controller. Beyond the aforementioned applications, OTFTs were applied as gas sensors, showing promising performance. This particular use is quite obvious since the electronic mobility of electroactive organic materials is highly sensitive to the chemical environment surrounding them. In addition, molecular engineering approaches make sensitivity and the selectivity of these materials possible by meticulous adjustment of pendant groups, functionalization with molecular receptors or even integration with biomolecular systems. Altogether, these strategies widen the number of analytes to which OTFTs show sensitivity.

Given the prior historical overview, the present manuscript reviews the applications of OTFTs in gas sensors, providing a concise description of the working principle, follows that with examples of organic materials and their processing and ends with some up-to-date gas sensing applications.

## 2. Organic Thin-Film Transistors

### 2.1. Operating Principles

An OTFT has a layered design consisting of an organic semiconductor (generally hole conducting/ p-type) thin film; a dielectric film; and three electrodes, namely, the source (S), drain (D) and gate (G). Depending on the position of these electrodes with respect to the semiconductor film, there are four possible transistor configurations, as shown in Figure 2. Among the possible combinations, bottom gate (BG) structures are the most interesting for gas sensing, since they expose the organic semiconductor to the target analytes. Top gate (TG) structures, on the other hand, are suitable for circuit fabrication (e.g., amplifiers and switches), as the semiconductor is encapsulated by the upper films. Independently, the source and drain electrodes are responsible for injecting and extracting the charge carriers, respectively, from the active layer, while the gate is separated from the semiconductor through a dielectric film of thickness $x_{i}$. The semiconducting channel is determined by just two parameters, width (*W*) and length (*L*), since charge transport occurs in a thin charge-accumulated layer of the semiconductor at the semiconductor/dielectric interface. The channel dimensions are limited by the presence of a gate electrode to tune its conductivity, as illustrated in Figure 3a.

A field-effect transistor has three operating modes, depending upon: (*i*) the gate-to-source voltage ($V_{GS}$) with respect to the threshold voltage ($V_{T}$); and (*ii*) the drain-to-source voltage ($V_{DS}$) with respect to the overdrive voltage ($V_{OV}=V_{GS}-V_{T}$). In addition, there are two types of FETs: a p-type, in which holes are the charge carriers in the channel, and an n-type, in which the charge carriers are electrons. Ambipolar materials may show both operating modes depending on the charge trapping and injection barriers. The operating modes for a p-FET device will be described in the following. Note that all voltages and channel current are negative for a p-FET.

The first operating mode, i.e., the cut-off, as shown in Figure 3b, is defined for $V_{GS}>V_{T}$ (i.e., $V_{OV}>0$ V), in which the applied $V_{GS}$ is not able to form a conducting path between source and drain, the channel is depleted of holes and the drain-to-source current ($I_{D}$) is zero. Ideally, the leakage current through the gate dielectric is also considered to be zero.

As $V_{GS}$ is shifted towards negative values lower than $V_{T}$ (i.e., $V_{OV}<0$ V), holes are accumulated at the dielectric/semiconductor interface. As shown in Figure 3c, an almost uniform charge distribution along the channel is achieved for $|V_{DS}\left|\ll \right|V_{OV}|$. Under these biasing conditions (usually, $|V_{OV}|$ greater than 10 times $|V_{DS}|$), $I_{D}$ will be different from zero and linearly dependent on $V_{DS}$ according to Equation (Equation 1).
(1)$$I_{D}=\mu C_{i}\left(W/L\right)\left(V_{GS}-V_{T}\right)V_{DS}$$

$C_{i}=\varepsilon_{i}/x_{i}$ is the gate capacitance density, whereas $\varepsilon_{i}$ stands for the gate dielectric permittivity. Note that the device behaves like a resistor with a conductivity of $\mu C_{i}$(*W*/*L*)($V_{GS}-V_{T}$).

In this operating mode, the OFET is also said to behave as a voltage-controlled current source equal to $g_{m}V_{GS}$. The transconductance ($g_{m}$) relates a change in output $I_{D}$ to an input $V_{GS}$ change with constant $V_{DS}$ bias:(2)$$g_{m}=\frac{\partial I_{D}}{\partial V_{GS}}=\mu C_{i}\left(W/L\right)V_{DS}$$

For $V_{DS}$ approaching $V_{OV}$, the charge carrier concentration is not uniform anymore, decreasing when moving towards the drain contact along the channel. It means that $V_{GD}$ is less negative than $V_{GS}$, and consequently, the charge accumulated around the drain is lower than that at the source. $I_{D}$ changes from a linear to a parabolic dependence on $V_{DS}$ according to:(3)$$I_{D}=\mu C_{i}\left(W/L\right)V_{DS}\left[\left(V_{GS}-V_{T}\right)-V_{DS}/2\right]$$

As shown in Figure 3d, at $V_{DS}=V_{OV}$, the channel is pinched-off closer to the drain electrode and the current in the channel equals:(4)$$I_{D}=\frac{1}{2}\mu C_{i}\left(W/L\right){\left(V_{GS}-V_{T}\right)}^{2}$$

Decreasing $V_{DS}$ to values more negative than $V_{OV}$ does not correspond to an increase in $I_{D}$. As the current remains constant, the device is said to be saturated. A similar reasoning can be extended to n-FETs, except for voltages and currents that have an opposite sign. There are also two subtypes of FETs, depending on the value of $I_{D}$ at zero $V_{GS}$. In an enhancement-mode FET, the current is very low and considered to be zero at $V_{GS}=0$ V. On the other hand, a depletion-mode FET can conduct current at this voltage. Additionally, it requires a $V_{GS}$ of opposite polarity to be cut-off (e.g., a positive $V_{GS}$ in a p-FET that normally operates under negative bias). More details on the equations given in this section are found in [32,33,34,35,36].

### 2.2. Characteristic Parameters

The operating modes described in the previous section are related to the p-type FET current versus voltage characteristic curves in Figure 4. For a deeper understanding of these curves, it is necessary to introduce the main parameters that are usually used to monitor transistor’s performance as a gas sensor.

#### 2.2.1. Charge Carrier Mobility

By definition, charge carrier mobility (μ) is described as the average charge carrier drift velocity ($v_{drift}$) per unit of electric field along the channel:(5)$$\mu =v_{drift}/E_{DS}$$

It is often translated as a measure of how efficiently charge carriers move along the conducting channel [32,34,36,37,38]. It is directly related to $g_{m}$, i.e., the slope of $I_{D}$ versus $V_{GS}$ in Figure 4b at $|V_{DS}\left|\ll \right|V_{OV}|$:(6)$$\mu_{lin}=\frac{g_{m}}{C_{i}\left(W/L\right)\left(V_{DS}\right)}$$

**Figure 4 materials-14-00003-f004:**
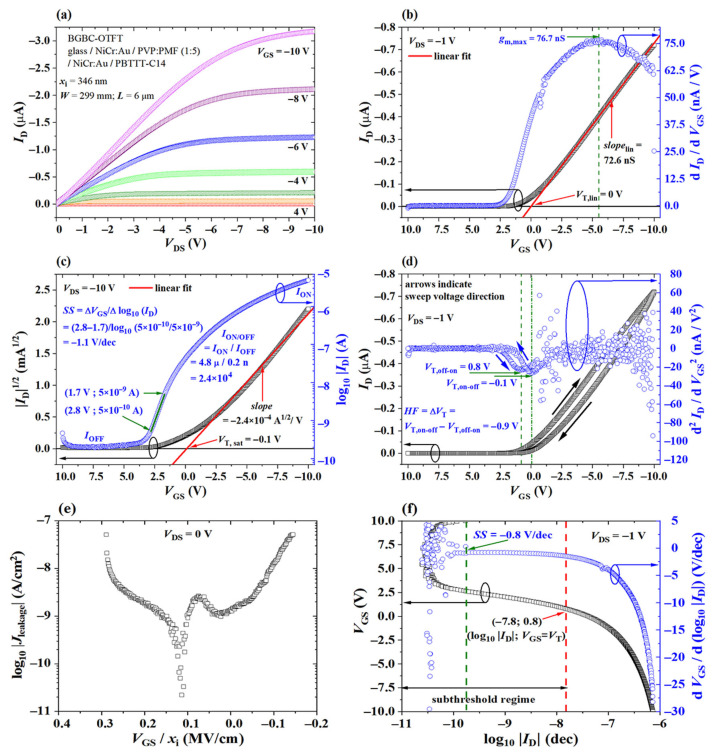
P-type FET characteristic curves: (**a**) $I_{D}$ versus $V_{DS}$ for $V_{GS}$ from 4 to −10 V. (**b**) Left axis: $I_{D}$ versus $V_{GS}$ for $V_{DS}=-1$ V to illustrate the linear fit for μ and $V_{T}$ calculation in triode operation. Right axis: $g_{m}$ versus $V_{GS}$ for μ calculation from $g_{m,max}$. (**c**) Left axis: $\sqrt{I_{D}}$ versus $V_{GS}$ for $V_{DS}=-10$ V to illustrate the linear fit for μ and $V_{T}$ calculation in saturation. Right axis: $I_{D}$ versus $V_{GS}$ in logarithmic scale to extract $I_{ON}$ and $I_{OFF}$, and calculate an approximate value for SS. (**d**) Left axis: off-to-on and on-to-off $I_{D}$ versus $V_{GS}$ scans for $V_{DS}=-1$ V featuring hysteresis. Right axis: plot of the second derivative of $I_{D}$ with respect to $V_{GS}$ to extract $V_{T}$ and illustrate the hysteresis factor calculation. (**e**) $J_{leakage}$ versus the perpendicular electric field in the channel ($V_{GS}$/$x_{i}$) for $V_{DS}=0$ V. (**f**) Plot of $V_{GS}$ versus log${}_{10}\left|I_{D}\right|$ and its first derivative to illustrate SS calculation. All data were extracted from [39].

Note that μ is also related to the slope of $I_{D}$ versus $V_{DS}$ in Figure 4a under similar biasing conditions. In addition, μ is not necessarily constant, so these slopes will not be constant either [40,41,42]. That is one reason why, in most cases, a linear fit of the curve $I_{D}$ versus $V_{DS}$ is performed in order to extract an average mobility value or $g_{m,max}$ is calculated to provide a maximum mobility value. It has been shown that μ in an OFET is dependent on the overdrive voltage by a power factor (γ) [32,34,37,38], according to:(7)$$\mu \left(V_{GS}\right)=k{\left(V_{GS}-V_{T}\right)}^{\gamma }$$
where *k* is a constant mobility value. The parameter γ is usually lower than one and depends on the conduction mechanism of the device, doping density and the dielectric permittivity of the active material. Other factors that strongly affect the mobility are grain size and intergrain defects, which in turn will depend on the surface prior to the semiconductor thin-film formation and the deposition parameters [43,44,45,46]. It is also worth mentioning that thinner dielectrics, lower *L* and larger μ values relate to larger switching speeds in digital circuits, but not necessarily to faster sensing responses and enhanced sensitivity [47].

As $V_{DS}$ approaches $V_{OV}$ in Figure 4a, the slope of the current versus voltage curve tends to zero and the FET is said to operate in saturation. A similar phenomenon happens as $V_{GS}$ approaches $V_{T}$ in Figure 4b. Since there is a region of the channel close to the drain contact that is depleted of carriers, it is not advised to relate that slope to μ. However, a rough estimation is obtained by a linear fit from the $\sqrt{I_{D}}$ versus $V_{GS}$ plot in Figure 4c:(8)$$\mu_{sat}=\frac{2}{C_{i}\left(W/L\right)}{\left(\frac{\partial \sqrt{I_{D}}}{\partial V_{GS}}\right)}^{2}$$

Note that a voltage scan at the desired operating mode is necessary in order to extract μ. Charge injection barriers and higher disorder in the semiconductor film at the electrodes in a bottom-gate/bottom-contact (BGBC) TFT are key factors that degrade the slope of the current versus voltage in triode mode. Those lead to bell-shaped curves at low $V_{DS}$ and an underestimation of μ. Even more precise models require the addition of $V_{OV}$-dependent resistance in series with the source and drain electrodes [33,35,38,45,48]. Although researchers seek to minimize its value through surface treatments, it is rarely used as a parameter to monitor gas sensing performance.

#### 2.2.2. Threshold Voltage

The $I_{D}$ current in both Figure 4a,b moves away from zero only for $V_{GS}$ more negative than the threshold voltage. In other words, $V_{T}$ is the minimum gate-to-source voltage required for accumulating charge carriers at the semiconductor/dielectric interface and forming a conducting path between source and drain electrodes [32,36]. It shows a strong dependence on the doping concentration, dielectric constant of the insulator, channel length and thicknesses of the active ($x_{s}$) and dielectric ($x_{i}$) layers [32,36]. In general, $V_{T}$ is desired to be as closest as possible to zero. That is often translated into a low operating voltage, and consequently, low power consumption, which is, thereby, beneficial to portable devices. A few strategies to increase gate capacitance are decreasing the thickness of the dielectric film or increasing its dielectric constant [49,50,51,52,53]. Since the threshold voltage relates to the charge accumulated at the semiconductor, its value is often assigned to the $V_{GS}$ value at the transition from a minimum (depletion) to a maximum (accumulation) gate-to-source capacitance ($C_{GS}$) [51,54]. From the accumulation capacitance, it is possible to calculate the gate dielectric capacitance $C_{i}$ (and, consequently, its thickness or dielectric constant). A fast, although dispersive, method to determine $V_{T}$ is from the intercept with $V_{GS}$ of the linear fit used to estimate μ in Figure 4b,c. Alternatively, $V_{T}$ can be related to a predetermined current level, which depends on the technology (i.e., semiconducting channel conductivity and dimensions), or more precisely, by the minimum of the second derivative of $I_{D}$ with respect to $V_{GS}$ shown in Figure 4d [32,34,45,55,56].

Hysteresis in the OFET characteristics is also illustrated in Figure 4d. That is usually translated into a hysteresis factor (HF) and quantified as a shift in the threshold voltage according to Equation (Equation 9) [37,51,57,58]. A shift in the transistor characteristics depending on voltage scanning parameters is not desirable. It is usually a sign that the device suffers from bias stress [43,59]. Since a shift in $V_{T}$ should be related to the gas sensor response, charge trapping in the films must be reduced. In other words, the semiconductor/dielectric interface must be improved through surface treatments. Alternatively, the stress could be minimized by biasing the sensor at voltages as low as possible in absolute values, but also for shorter times and not so frequently. Compared to mobility calculations, determining $V_{T}$ usually demands wider voltage scans from cut-off to saturation or triode. Monitoring the capacitance requires even more expensive hardware and more design complexity.
(9)$$HF=\Delta V_{T}$$

#### 2.2.3. Current On/Off Ratio

The ratio of current in accumulation and depletion modes is known as the on/off ratio [36]. The off current depends on the channel conductivity ($\sigma_{s}$) and dimensions by:(10)$$I_{OFF}=\sigma_{s}x_{s}\frac{W}{L}V_{DS}$$

For a device switching from cut-off to saturation, the on/off current ratio is expressed as:(11)$$I_{ON/OFF}=\frac{\mu C_{i}{\left(V_{GS}-V_{T}\right)}^{2}}{\sigma_{s}x_{s}V_{DS}}$$

This parameter is graphically extracted from $I_{D}$ versus $V_{GS}$ plots in logarithmic scale, as shown in Figure 4c. A high dielectric constant and a thin dielectric film are needed to increase $C_{i}$. That usually leads to an increase in the leakage current density ($J_{leakage}$) from the gate electrode, which negatively impacts in $I_{OFF}$ [49,50,51,52,53]. $J_{leakage}$, as shown in Figure 4e, should be minimized so the sensor can operate at low biasing voltages. Other key factors in generating a large current modulation demand thin and low doped semiconducting films [60]. Whereas a high on/off ratio approaching $10^{8}$ is an essential requirement for display applications [61,62,63,64], this is not a mandatory condition for sensors. In addition, similar to the threshold voltage, wider voltage scans from cut-off to saturation modes are necessary to estimate $I_{ON/OFF}$.

#### 2.2.4. Subthreshold Slope

The subthreshold slope (SS) is the variation in the gate biasing to produce one decade change in the drain current [34,36,55,65,66,67,68]. A graphical estimation from $I_{D}$ versus $V_{GS}$ plots in logarithmic scale is illustrated in Figure 4c. A more precise calculation involves plotting $V_{GS}$ as a function of log${}_{10}\left|I_{D}\right|$ and its first derivative, as shown in Figure 4f. SS is determined to be the minimum of $\partial V_{GS}$/$\partial log_{10}\left(I_{D}\right)$ in absolute values for $V_{GS}$ > $V_{T}$. It depends on the impurity concentration, interface state and trap density, as for $V_{T}$ and $I_{ON/OFF}$. The exponential increase in channel current in absolute values seen in Figure 4c is a direct consequence of the transition from depletion to accumulation of charge carriers. In an OTFT, SS is closely related to the mobility enhancement for carrier hopping. A lower trap density (e.g., single crystal FETs) is desirable to achieve steeper slopes, and consequently, better switching behavior [51,55,65,66,67,68,69]. A low SS in absolute values is frequently associated with a sensor that suffers less from bias stress and operates at lower voltages. However, it is rarely used as a sensitivity parameter. The interaction between trapping sites and gaseous analytes is usually monitored through mobility or threshold voltage shifts [70,71,72,73].

#### 2.2.5. Gas Sensing Response

The performance of a gas sensor is characterized by the response (*R*, also named responsivity) of a TFT parameter (e.g., $I_{ON}$, μ, $V_{T}$) at a specific voltage bias (*V*) as:(12)$$R=\frac{\Delta X}{X_{0}}$$
in which ΔX is the difference between the parameter extracted at a specific gaseous analyte concentration (*c*) and its value at a reference gas (e.g., atmospheric air or an inert gas), in which c=0 ppm. Sensitivity (*S*), on the other hand, is defined as:(13)$$S=\frac{dR}{dc}$$

This gas sensing parameter is usually approximated by the slope from a linear fit of the *R* versus *c* plot within a specific concentration interval. In addition, the intercept of that line with the concentration axis with zero response is often presented as the limit of detection (LoD). Due to the presence of noise during measurements, a better estimated value is given by the intersect with at least twice the noise-floor level. Finally, since the sensor response does not occur immediately, there are two further parameters related to timing. The onset time ($t_{set}$) is the interval needed for ΔX to vary from 10% to 90% of its value. The reset time ($t_{reset}$) is calculated in the same way, except that *X* varies from its value at a specific *c* to its value at the reference atmosphere. Although hardly provided as a separate subsection, more details on gas sensing parameters can be found in previous literature reviews [74,75,76,77,78,79].

### 2.3. Fabrication Techniques

Nowadays, most OTFTs for gas sensing applications use a combination of well-established thin-film formation techniques and recently-developed processes for flexible substrates. Since 2015, most gas sensors were demonstrated in a bottom gate structure over thermally-oxidized Si wafers and by employing well-known clean-room fabrication techniques [70,72,73,80,81,82,83,84,85,86,87,88,89,90,91,92,93,94,95,96,97]. Physical vapor deposition of metals (e.g., thermal evaporation, electron-beam PVD and sputtering) [70,72,73,80,82,83,84,85,86,87,88,89,91,94,95,96,97,98,99,100,101,102,103,104,105,106,107] and photolithography [70,93,99] were used for electrode thin-film formation and patterning, respectively. Semiconductor deposition usually involves one of the two following techniques: (i) spin coating [70,73,81,82,88,89,90,94,96,98,99,100,101,104,105,105] and (ii) thermal evaporation [83,103,107]. Spin coating is widely used for the formation of photoresist layers during lithography. It involves casting an organic polymer dissolved in organic solvents onto a substrate either at zero rpm or already rotating. After being rotated at a predefined limit speed (~500–6000 rpm) for a predefined amount of time (~10 s–2 min), the thin film is dried either on a hot plate or inside an oven. In the case of crystalline semiconductors, this last step, often called thermal annealing, has also the role of promoting thin-film crystallinity [108]. The spin coating technique is illustrated in Figure 5a. Thin-film thickness depends on rotation speed and solution properties, such as concentration and viscosity [109]. Surface treatments prior to wet processing are often used to improve uniformity and coverage. A pre-patterned film exposing areas with different surface energies can be used to achieve a patterned film after spinning. Another widely-used deposition process, thermal evaporation requires heating of an organic material to allow it to either evaporate or sublimate. The organic material must be placed inside an electrically heated crucible, and the evaporation/sublimation happens under high-vacuum conditions (~10${}^{-7}$–10${}^{-5}$ Torr). A substrate is positioned upside down and on top of the crucible at a distance of a few tens of centimeters [110]. Thermal evaporation is shown in Figure 5b. In this case, patterning is achieved by the use of stencil or shadow masks positioned as close as possible to the top surface of the sample. Thin-film thickness is controlled by the evaporation rate. Among the major drawbacks of both techniques are the production of a large amount of waste (i.e., low yield) and are challenging to adapt to large-area substrates.

Throughout the years, new deposition techniques were developed in order to promote crystallinity, patterning organic films and processing over large area substrates. Printing of organic materials was inspired by ink-jet printers found in almost all offices in the past decades. Among its positive characteristics are the additive patterning and reduced solution waste. In other words, droplets of the solution containing the desired material are added on the substrate exactly where needed. These droplets are ejected from the nozzle due to a volume expansion either in the liquid by a heating electrical resistance or in the nozzle by a piezoelectric film [111]. Ink-jet printing from piezoelectric-based nozzle and cartridge is illustrated in Figure 6a. One of its major drawbacks is resolution, usually limited to ca. 1 μm. That means that the minimum separation between source and drain electrodes can not be smaller than the resolution, which is critical for fast circuits, but not necessarily for gas sensors. Developing a thin-film with desired properties (e.g., thickness, roughness and resolution) requires tuning the parameters for droplet formation, such as solution viscosity and surface tension, printing speed and cartridge temperature [111]. Wet-processed single crystals are possible by printing anti-solvent droplets before the solution ink [112]. Other techniques were also ameliorated throughout the years to enhance thin-film crystallinity. Meniscus-guided coating (MGC), which involves the evolution of a solution meniscus acting as an air/liquid interface for solvent evaporation, is one of them. The meniscus is usually formed by a reservoir containing the solution (e.g., the tip of a syringe or a slot) in close proximity with a solid surface (80–100 μm). Due to relative movement between the substrate and the solution reservoir, the meniscus is dragged along the substrate surface, the solvent evaporates, the solute precipitates at supersaturation and a thin-film is formed [109]. Blade coatings/doctor blades, also named bars or knives coating depending on blade geometry, slot die, solution shearing and pneumatic nozzle printing are based in a similar principle, since a small volume of the solution is cast on a surface and slowly spread to carefully control film crystallization [113]. Tuning single crystal dimension and growth direction usually involves micro-structuring the blade surface in contact with the solution [114]. Solution shearing with a micro-structured blade, as an example of MGC, is illustrated in Figure 6b.

**Figure 6 materials-14-00003-f006:**
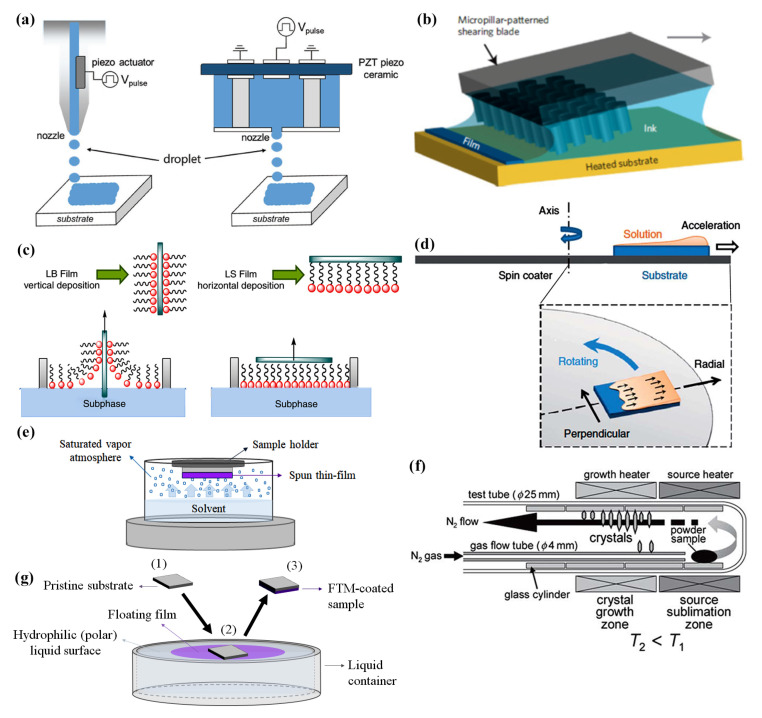
Illustration of currently used and innovative techniques for thin-film deposition of organic transistors as gas sensors: (**a**) Ink-jet printing. Reprinted from [111], with permission, from WILEY-VCH Verlag GmbH and Co. KGaA, Weinheim, ©2019. (**b**) Solution shearing. Reprinted by permission from Springer Nature Customer Service Centre GmbH: Nature, Nature Materials [114], ©2013. (**c**) Langmuir-based monolayers. Reprinted from [115], copyright 2017, with permission from Elsevier. (**d**) Off-center spin coating. Reprinted by permission from Springer Nature Customer Service Centre GmbH: Nature, Nature Communications [116], ©2014. (**e**) Solvent vapor annealing; (**f**) Physical vapor transport. Reprinted from [117], copyright 2008, with permission from Elsevier. (**g**) Floating film transfer method.

Dip coating, which can also be classified as MGC, is a fast, low-cost and straightforward technique. It is a three-step process: (1) a substrate is dipped into a solution; (2) a wet film is formed after its immersion; and (3) the solvent is evaporated [91]. This principle has inspired other thin-film formation processes, such as electrostatic self-assembly (ESA), in which the process parameters are more carefully adjusted for thickness control. Mostly used for biosensors, ESA involves two solutions of differently charged species (i.e., anionic and cationic solutions), so that electrostatic attraction favors the formation of a bilayered-structured film [118,119].

In other techniques derived from dip coating, such as Langmuir–Blodgett (LB) and Langmuir– Shäfer (LS), there is an even more careful control of uniformity and thickness to achieve the formation of just one molecular layer (also called a monolayer) [115]. Specifically, LB involves the coordinated movement of mobile barriers that compress the molecules spread at the air–water interface, and a device that moves the substrate perpendicularly to the solution. In LS, which is less common than LB due to a worse surface coverage, a monolayer is deposited on the substrate horizontally with respect to the liquid surface [115]. LS of amphiphilic molecules is shown in Figure 6c. In this case, the molecule self-organizes at the surface with the polar head group (hydrophilic) into the polar liquid (e.g., water) and a non-polar tail part (hydrophobic) in the air. Thicker films are generated by repeating the deposition process and stacking more monolayers on top of each other. Self-assembled monolayers (SAM) can also be formed by chemical reactions with atoms at the surface of a film or substrate after immersion into a solution containing the molecule to be deposited [120]. Usually, the surface needs to be activated by O${}_{2}$ plasma, UV-ozone or a strongly oxidizing acid. Alternatively, SAM formation can occur under gas flow in a saturated vapor environment.

Although spin coating is still used, a few adaptations have been promoted in order to enhance the film’s crystallinity. In off-center spin coating, the geometric center of the substrate is placed outside of the axis of rotation, so that the film spreads along one specific direction, instead of radially [116]. Off-center spin coating is illustrated in Figure 6d. Higher crystallinity is usually achieved by lower spinning speeds and mixing with less-volatile solvents. Alternatively, the formation of large crystals (> 100 μm) can be induced by post-deposition solvent vapor annealing (SVA) [121]. Once the thin-film is exposed for a few hours to the saturated vapor of a volatile organic solvent (e.g., chloroform), solvent molecules are adsorbed onto the film’s surface and allowed to diffuse into it. By lowering the glass transition temperature and viscosity of the material, partially dissolved molecules in the film are allowed to rearrange [122]. SVA, illustrated in Figure 6e, is similar to slow drying (SD) [95], since the idea is to promote crystallinity by allowing the solvent to evaporate slowly from the film. Not only office printers were adapted for organic electronics. Spray coaters used, for instance, in painting by the automotive industry are also used to deposit organic thin-films for electronics [106]. Among the available strategies to atomize the ink, pneumatic-based systems are widely-used. In the process, pressurized gas (e.g., N${}_{2}$) breaks up the liquid into droplets, which are then carried towards a substrate fixed at a pre-defined distance from the nozzle’s tip. Thin-film properties will depend on the solution conditions (surface tension, concentration and viscosity), and on the gas flow and the nozzle geometry [123,124].

Although partially losing the benefits from wet processing, organic semiconducting single crystals can be grown by physical vapor transport, and then, transferred to a substrate for OTFT fabrication [92]. In this technique, the source material is placed near one extreme of a quartz or glass tube and the material is heated to its sublimation temperature or above under the flow of an inert gas (e.g., N${}_{2}$, Ar or He). Once the organic semiconductor is vaporized, it is carried down the tube by the inert gas, where it re-solidifies due to the presence of a temperature gradient. Physical vapor transport is illustrated in Figure 6f. These crystals typically vary in size from tens of nanometers to several micrometers in thickness, while forming even centimeter-long plates [125]. Films can also be grown on one substrate or at the surface of a liquid, and then, transferred to a different substrate. For instance, polymer films can form from a drop of a hydrophobic polymeric solution over a viscous hydrophilic liquid surface. As the solution spreads over the surface, the solvent evaporates rapidly, and at the same time, the viscous force of the liquid substrate acts against the spreading. A thin floating film of the polymer is obtained, which is then transferred to or stamped onto other substrates. The floating film is then fished from the solution similarly to the LS method [84,87,93,126,127]. This technique, known as floating film transfer method (FTM), is illustrated in Figure 6g. Graphene films grown by dry techniques as CVD can be separated from a pristine substrate by etching, and then, transferred to another substrate. A poly(methyl methacrylate) (PMMA) film under a poly(dimethyl siloxane) (PDMS) stamp is usually used as a sacrificial layer in the process, later released from the stamp by an organic solvent (e.g., chloroform) [128]. Liquid-bridge-mediated nanotransfer molding (LB-nTM) is a technique that integrates transfer and patterning of organic single crystals [80]. First, a flexible patterned cast containing the structures must be filled with the organic ink solution. After solidifying the ink, the cast with the organic structures is brought into contact with a substrate surface covered by a thin solvent layer. The solvent evaporates by a thermal treatment, and then, the solidified structures in the cast are transferred to specific positions on the target substrate. Although focusing on the development of photovoltaics, most of these innovative techniques have already been successfully adapted to roll-to-roll processing over large area substrates [129]. Finally, it is worth mentioning that organic dielectrics and conductors are mostly wet-processed polymers by these same techniques, except that, in general, uniformity (e.g., thickness, roughness and reduced aggregation) is much more important than crystallinity.

### 2.4. Organic Materials

The vast majority of organic semiconductors are p-type materials including aromatic units. In this group, there are polymers, such as poly (3-hexylthiophene) (P3HT) [70,73,80,81,90,94,98,105,106], poly(3,3${}^{\prime \prime \prime }$-didodecylquaterthiophene)(PQT-12) [87], poly(2,5-bis(3-tetradecylthiophen-2yl)thieno(3,2- b)thiophene) (PBTTT) [84], which are processed in solution, and small molecules, such as pentacene [103], copper phthalocyanine (CuPc) [107], dinaphtho [2,3-b:2${}^{\prime }$,3${}^{\prime }$-f] thieno [3,2-b] thiophene (DNTT) [83] and dinaphtho[3,4-d:30,40-d0]benzo[1,2-b:4,5-b0]dithiophene (Ph5T2) [92], which are thermally-evaporated in high vacuum conditions. Most conjugated small organic molecules exhibit poor solubility in organic solvents. Recently-synthesized small organic molecules, however, were made soluble by the addition of side chain and functional groups. A few examples are 2,7-dioctyl benzothieno[3,2-b]benzothiophene (C${}_{8}$-BTBT) [85], triethylsilylethynyl-anthradithiophene (TES-ADT) [97] and 6,13-bis(triisopropylsilylethynyl)-pentacene (TIPS-pentacene) [95,96,101,102]. In addition, small organic molecules can be modified to form self-assembled monolayers. Chen et al. increased the solubility of some phthalocyanines for quasi-LS deposition and tuned gas sensing selectivity at the same time by introducing a different number of thiophenoxy groups at the macrocycle periphery. In addition, bis(phthalocyaninato) of rare earth metals (e.g., Eu[Pc(SPh)${}_{8}$]${}_{2}$) featured ambipolar behavior [86]. Sizov et al. synthesized 1,3-bis11-(7-hexyl[1]benzothieno[3,2- b][1]benzothien-2-yl)undecyl-1,1,3,3-tetramethyldisiloxane (O(Si-Und-BTBT-Hex)${}_{2}$), an organosilicon derivative of [1]benzothieno[3,2-b][1]-benzothiophene (BTBT), for LS deposition over SiO${}_{2}$ [72]. Block copolymers have also offered the opportunity of new organic semiconductors. For example, Wei et al. deposited semiconducting helical nanofibrils for gas sensing applications based on a p-type copolymer made of poly(4-iso-cyano-benzoic acid 5-(2-dimethylamino-ethoxy)-2-nitro-benzylester) and P3HT (PPI(-DMAENBA)-b-P3HT) [82]. Copolymers are also amenable to build solution-processed, ambipolar, stable semiconductors. The motivations for development of ambipolar materials stems from the need to integrate p-n junctions within a single backbone, which decreases the bandgap and thus allows for a more efficient light-harvesting in photovoltaics. These copolymers are composed of alternating electron-rich (donor) and electron-deficient (acceptor) units, also called D–A moieties. A trending approach is to integrate thiophene or bithiophene rings with electron-accepting units, such as diketopyrrolopyrrole (DPP) [85,93,99], bis(2-oxoindolin-3-ylidene)-benzodifuran-dione (BIBDF) [89], isoindigo [88] and benzothiadiazole (BT) [91,100,104]. Gold [70,72,73,81,82,83,84,85,86,87,88,89,90,91,92,93,94,95,97,98,99,100,101,102,103,104,105,106,107] and silver [80] are common choices for sourcing and draining electrodes, since charge conduction usually happens at ca. 5 eV in these organic materials. The chemical structures of the currently most-used semiconducting molecules for gas sensing applications from OTFTs are shown in Figure 7.

Although many gas sensors are formed on Si wafers, the interface of an organic semiconductor with an inorganic oxide usually features a high density of traps, which leads to high subthreshold slopes and hysteresis in OTFTs. Prior to applications in organic electronics, surface treatments were already used in lithography, targeting the removal of moisture and consequently, the promotion of an increased adhesion of photoresist onto wafers after spin coating. In OTFTs, however, the surface needs to be first activated by an oxidizing treatment. Then, dangling bonds and hydroxyl groups at the surface of an inorganic oxide are replaced by carbon-based molecules. Hexamethyldisilazane (HMDS) [70,86] and octadecyltrimethoxysilane (OTS) [72,85,87,88,91,93] are largely used for SAM treatments. Alternatively, a thin organic dielectric buffer layer (<100 nm) can be stacked over the inorganic oxide [82,83,84,85,89,95,98,107]. In such a process, however, the challenge of stacking two organic molecules comes up. In this case, it makes sense to directly look for organic substrates and dielectric substitutes. Replacement of highly-doped Si wafers, which are used as a common gate electrode, usually leads to the utilization of either aluminum [99,100,104] or indium-doped tin oxide (ITO) [91,96,98,101,102,103,105,107]. The latter, a transparent conductor, opens up the possibility for the fabrication of optically transparent devices on glass. Conversely, mechanically-flexible sensors require polymeric substrates, such as poly(ethylene 2,6-naphthalate) (PEN) [100,106] and poly(ethylene terephthalate) (PET) [85,104].

The migration to alternative platforms for device processing, however, makes it necessary to find a replacement also for SiO${}_{2}$. This high-quality dielectric oxide is rather inconvenient for fabrication of low-cost flexible sensors, since it is grown by thermal oxidation of Si at a temperature (>$1000\;{}^{\circ }C$) well-beyond melting of flexible substrates (200–$400\;{}^{\circ }C$). Inspiration came from wet-processed polymer dielectrics already used in clean room facilities for the fabrication of integrated circuits (IC) and microelectromechanical systems (MEMS) on Si wafers. Among the most-used insulating polymers one can cite the fluoropolymer Cytop${}^{\mathrm{TM}}$ [82,84,89,89], as well PMMA [96,98,99,101,102,104,105,106], poly(imide) (PI) [100], poly(4-vinylphenol) (PVP) [85], poly(styrene) (PS) [103,107] and poly(vinyl alcohol) (PVA) [107]. Kapton${}^{\mathrm{TM}}$, a PI-based dielectric, withstands both high temperatures (<$400\;{}^{\circ }C$) and organic solvents, and for that reason, it is often used as a substrate. Since most dielectric polymers are not chemically-resistant, cross-linkers (e.g., 4,4${}^{\prime }$-(hexafluoroisopropylidene)- diphthalic anhydride (HDA) [85] and poly(melamine-co- formaldehyde) methylated (PMF) [39]) are blended to the dielectric solution before deposition. Even though most electrodes are based on noble metals, it is worth mentioning that compounds based on poly(aniline) (PAni), poly(3,4-ethylenedioxythiophene):poly(styrene sulfonate) (PEDOT:PSS) and graphene are alternatives to achieve all-organic sensors. They are not only characterized by a reasonable optical transparency, similarly to organic dielectrics and substrate, but also for lower contact barriers with organic semiconductors for charge transport. It is not a surprise that these polymeric conductors are often used as thin (25–100 nm) hole injection layers (HILs) in OLEDs and hole transport layers (HTLs) in OPVs [130,131,132,133]. The chemical structures of the recently most-used organic dielectrics, cross-linkers, SAM, substrates and conducting molecules for gas sensing applications with OTFTs since 2015 are shown in Figure 8.

As an additional point, wet-processing allows for blending organic semiconductors with other materials. Owing to the phase segregation that occurs after solvent evaporation, insulating polymers tend to deposit at the semiconductor/bottom gate dielectric film interface. Recent works have shown organic semiconductors blended to PS [95,98]. First, it acts as a surface treatment to passivate traps at the dielectric/semiconductor interface. In addition, it tends to encapsulate the dielectric to prevent $V_{T}$ shifts in gas sensing experiments. In some cases, it can even increase the surface area and improve gas penetration in the semiconducting film. That is actually the case of porogenic materials. For instance, Besar et al. mixed PBIBDF-BT with N-(tert-butoxy-carbonyloxy)-phthalimide to prepare a porous film, since the latter decomposes above $150\;{}^{\circ }C$ [90]. Wu et al. added poly(1,4-butylene adipate) (PBA) [89] in order to prepare the macroporous semiconductor film by washing-off the additive. Even n-type small organic molecules and dielectric polymers can find application as a porogenic material. Park et al. blended phenyl-C${}_{61}$-butyric acid methyl ester (PCBM) to P3HT [73]. For selective etching of PCBM, thin films were immersed in a bath of n-butyl acetate (BA), followed by spin-drying. Lu et al. evaporated DNTT over PS microspheres. A porous film of the semiconductor was easily obtained by taping the spheres out of the substrate [83]. Literature has also described blends of p-type semiconductor polymers (e.g., poly(9-vinylcarbazole) (PVK) [105]) and n-type inorganic nanoparticles (e.g., CdSe [87] and ZnO [94]), and with metal particles (e.g., Pd [90]). In all these cases, charge transport, and consequently, the device current, are affected by the presence of materials of different conductivities or the formation of p-n junctions along the transistor channel. Finally, stacked evaporated films can alter sensor selectivity (e.g., Ph5T2 on CuPc single crystal nanowires (NWs) [92]). The chemical structures of the abovementioned porogenic molecules for gas sensing applications in OTFTs are given in Figure 8.

### 2.5. Gas Sensing Applications

The electrical conductivity in organic semiconductors is a sum of contributions from charge transport along a single molecule; between molecules; and finally, between grains. As mentioned earlier, organic semiconductors share a common molecular structure endowed by a conjugated network of single and double bonds. Since single bonds are longer than double bonds, a band gap opens-up between HOMO and LUMO orbitals, which are equivalent to valence and conduction bands in conventional inorganic semiconductors. For trans-poly(acetylene), there is an electronic energy level (state) in the mid-gap called soliton. From a chemistry point of view, the soliton is a neutral radical species that can move along the polymer backbone, back and forth, without losing its energy. Further doping, by either addition or removal of electrons, leads to charged solitons, which can indeed conduct electrical current [134]. For aromatic semiconducting polymers after doping, electron-phonon coupling produces polarons, which are radical cation or radical anion species populating the gap and are responsible for electrical conduction along the polymer backbone [135]. Graphene exhibits no gap, since pi-bonding and pi-antibonding orbitals meet in the same point, although it still shows polaronic carriers [136]. Nonetheless, defective graphene (e.g., graphene oxide, reduced graphene oxide, heteroatom substituted graphene) does have a set of mid-gaps, which extend from UV to near-infrared [137]. Carbon nanotubes may behave between semiconductor and metal, depending upon the chirality of the tubes (i.e., the way a hypothetic graphene sheet is rolled up to form the tube) and the conductivity is due to delocalized pi-electrons [138]. In a second level, conductivity relies on hopping of charge carriers between alike polymeric chains, sheets and tubes, which is temperature-dependent (Arrhenius-type).

All these modes are sensitive to the chemical environment where the organic semiconductor is confined. For the gas sensing purposes, a general mechanism of detection establishes that the molecule of a chosen analyte may interact with the charge carriers within the semiconductor sensing layer by either charge-transfer (donating or accepting) or polarization. Since most of organic semiconductors behave as p-type, electron-donating molecules nearby will decrease the charge-carrier density, thereby increasing the resistivity of the sensing layer. On the other hand, electron-withdrawing molecules will increase the charge-carrier density, consequently decreasing the resistivity of the sensing layer. Alternatively, the analyte molecule may solely cause polarization of the sensing layer, which is detected by a capacitive current. Obviously, the prevalence of each mechanism may be modulated by the operational conditions, especially voltage bias and temperature, since charge-carrier density and analyte adsorption are extremely dependent on these variables [139,140,141,142].

When compared to their inorganic counterparts, a significant advantage of organic semiconductors is the possibility to impart them with selectivity by means of molecular engineering approaches [83,93]. In addition, these semiconductors have electronic and mechanical properties that make them great players for the development of low-cost flexible electronics [143]. Nonetheless, they are more susceptible to environmentally-driven degradation than inorganic semiconductors. For instance, oxygen and moisture present in the atmosphere not only influence their charge-transport properties to several orders of magnitude, but may also contribute to irreversible damage. A way to overcome this problem is to process and encapsulate the devices under inert atmosphere inside glove-boxes before they are put in contact with real environmental conditions. Although this sensitivity is a bottleneck for OLEDs and OPVs, it is essential for gas sensors.

Mankind is often exposed to gases originating from chemical industries, water waste and environmental pollution. That opens up the way to use organic-based devices as gas sensors. Much more than that, gas sensors may play a unique role in several fields, such as in (i) the pharmaceutical industry, (ii) disease diagnosis, (iii) defense and safety, (iv) food deterioration and (v) environment monitoring.

It is worth noticing that there are several methods available for gas detection exhibiting selectivity and sensitivity, including gas chromatography (GC), mass spectrometry (MS) and optical chemical sensing [144,145,146,147,148,149,150,151,152]. Nonetheless, they are based on robust and costly equipment, which are hardly implemented at remote locations, and therefore, do not allow for on-site monitoring. Moreover, they require a high-level of expertise for their operation and data interpretation, which is usually time-consuming. In contrast, OTFT-based gas sensors may offer portable devices of easy operation, even for unskilled operators, which could perform real-time and on-site detection at much lower cost.

In 2000, Crone et al. demonstrated that OTFTs have suitable properties (e.g., sensitivity and reproducibility) for use in gas sensors [153]. Since then, several other organic materials and analytes have been proposed. Table 1 summarizes the most relevant OTFT-based gas sensors and respective performances, as reported since 2015. In these devices, the analyte adsorption and diffusion into the active material is converted into an electrical signal. For that, a change in the thin-film effective charge carrier mobility (μ) is translated into a change in the semiconducting channel conductivity. That interaction generates an amplified step in the drain current ($I_{D}$). Therefore, the organic transistor behaves simultaneously as transducer and amplifier of the gas sensing response. The main groups of analytes detected by OTFT-based gas sensors are atmospheric gases (e.g., N${}_{2}$, O${}_{2}$, CO${}_{2}$ and H${}_{2}$O), volatile organic compounds (VOCs) [70], poisonous and explosive gases (e.g., NH${}_{3}$, CO${}_{2}$, CO and H${}_{2}$S) [83]. Many VOCs, and many poisonous and explosive gases, are also compounds from the metabolites produced by living organisms (e.g., bacteria and cells) [154]. Gas sensing performance in response to widely-studied gases will be detailed in the following subsections.

#### 2.5.1. NH${}_{3}$

Ammonia is a dangerous gas, highly toxic to human beings and explosive under some conditions [155,156]. Under normal conditions of temperature and pressure, it is colorless and corrosive with a strong-smelling odor. However, NH${}_{3}$ is an abundant nitrogenous waste in the urine or feces of many animals [157]. In addition, degradation of organic matter usually produces NH${}_{3}$. It is not a surprise that ammonium nitrate (NH${}_{4}$NO${}_{3}$) is a widely-used fertilizer [158]. Unfortunately, it is highly explosive under heat. In order to detect NH${}_{3}$, several p-type organic semiconductors have been tested as the active layer in OTFTs [70,72,73,80,81,82,83,84,85,87,88,98,99,100]. Owing to its electronic structure, NH${}_{3}$ is a Lewis base and displays an electron-donating character, thereby reducing the hole mobility in p-type semiconductors. Additionally, it may act as a dedoping agent by compensating for an ambient oxidant (e.g., H${}_{2}$O and O${}_{2}$) [70].

Although highly-sensitive to changes in relative humidity levels [70,73], P3HT stands out in bottom-gate OTFTs for NH${}_{3}$ detection [70,73,80,81,82,98]. P3HT nanowires 100-nm wide deposited by liquid-bridge-mediated nanotransfer molding decreased LoD down to 8 ppb with 68.8% response at 1 ppm [80]. Wei et al., on the other hand, synthesized a block copolymer from P3HT to form 30–50 nm-wide nanofibrils and achieved similar results with a response time of just 3–7 s [82]. Most approaches to improve sensitivity involve microstructuring the active layer, and consequently, increasing the surface area. Lu et al. reported a porous OTFT based on thermally-evaporated DNTT over PS microspheres deposited over plasma-treated oxidized Si wafers. After removal of the spheres by taping them out of the substrate, Au top electrodes were evaporated through a shadow mask. At low concentration (tens of ppb), the porous device exhibited a sensitivity of 340 %/ppm. As a comparison, the pristine OTFT only exhibited a sensitivity of 20 %/ppm (almost 20 times lower than the porous devices) [83]. DNTT-based transistors still showed selectivity, although with a slower response time (95 s) compared to P3HT transistors. Among the best-performant NH${}_{3}$ gas sensors, Zhang et al. achieved a LoD lower than 1 ppb with a 27.8% response at 1 s (see Figure 9a) [85]. In order to do that, poly(diketopyrrolopyrrole- thiophene-thieno[3,2,b]thiophene-thiophene) (DPP2T-TT) was deposited by meniscus-guided printing on nanoporous cross-linked PVP with HDA over Si/SiO${}_{2}$ substrates. According to the authors, nanopores enabled direct access to highly reactive sites otherwise buried in the conductive channel. It originates from two aspects: (i) the p-type organic semiconducting molecules in the conducting channel are positively charged for hole transport; and (ii) the backbone of these polymers is oriented edge-on with respect to the substrate, thereby favorably exposing the π-electrons to the pore wall. When exposed to electron-donating gases such as NH${}_{3}$, n-type semiconductors exhibit a positive current response. Chen et al. observed this kind of behavior for europium (III) complexes [86]. Despite its ambipolar TFT behavior, the gas sensing response to NH${}_{3}$ was mostly due to the presence of peripheral electron-withdrawing thiophenoxy substituent. Exposure to the gas molecules acted to cancel this effect by a chemical doping.

As shown in Table 1 for NH${}_{3}$ detection, the limit of detection can range from 1 ppb to 1000 ppm, responsivity from 10 to 98.3% and response time from 1 to 180 s. Almost all OTFTs are p-type operated at a maximum voltage from −1 to −80 V, being highly selective to this gaseous analyte. These detection limits are better than what is usually obtained with surface acoustic wave (SAW) and electrochemical sensors; comparable to laser photoacoustic spectroscopy (LPAS) and metal-oxide (MOx) chemosensors; and worse than selected ion flow tube MS (SIFT-MS) and tunable diode laser absorption spectroscopy (TDLAS) [146,159].

#### 2.5.2. VOCs

Volatile organic compounds, or VOCs, are carbon-based chemical compounds that evaporate under normal atmospheric conditions from certain liquid or solid sources (e.g., paints and fuels) [145,160] and living organisms (e.g., plants and humans) [161,162,163]. Monitoring the presence of VOCs finds application in indoor air quality, environment pollution control [160], disease diagnosis [162,163] and explosives detection [145]. Among the most studied VOCs are hydrocarbons, aldehydes, alcohols, ketones and organohalogen compounds. Acetone in breath is a potential volatile biomarker for diagnosis of diabetes mellitus. Other ketones, including acetone itself, are potential biomarkers for early lung cancer detection [164]. Cavallari et al. integrated P3HT-based transistors and chemosensors for non-invasive disease diagnosis and environmental monitoring through the detection of gaseous analytes such as methanol, acetone and chloroform [70]. Devices responded, mostly, by conductivity increases due to attractive electrostatic interaction among polymer molecules by induced van der Waals force after adsorption of polar VOCs. The consequent reduced average spacing increases the available density of states for interchain polaron hopping. In addition, the authors suggested that using the multiparameter characteristic of TFTs enhanced electronic nose capabilities for VOC detection, since P3HT is also highly sensitive to NH${}_{3}$. Although hardly sensitive at the sub-ppm level, there are strategies to boost analyte concentration in a sample and overcome the limit of detection of the device for practical applications [165]. Higher selectivity towards a particular volatile organic compound usually demands the deposition of an additional sensing layer or mixing two or more materials in the active layer [90].

**Table 1 materials-14-00003-t001:** Summary of gas sensing performances from organic thin-film transistor (OTFT)-based devices since 2015.

Analyte	Semiconductor	Technique	Structure ‡	LoD	*R* (%)	$t_{\mathrm{set}}$ (s)	*V* (V)	Selective	Year	Ref.
Acetone	P3HT	spin coating	BGBC	440 ppm	1.5 @ 244 ppm ($I_{ON}$)	68	−1	No	2015	[70]
Chloroform	P3HT	spin coating	BGBC	1100 ppm	6 @ 222 ppm ($I_{ON}$)	201	−1	No	2015	[70]
Ethanol	P3HT (porous)	spin coating	BGTC	100 ppm	33.7 @ 1000 ppm (μ)	-	−60	Yes	2019	[73]
Ethylene	P3HT:Pd (porous)	spin coating	BGBC	25 ppm	30.2 @ 25 ppm ($I_{ON}$)	300	−60	Yes	2017	[90]
Formaldehyde	C${}_{8}$-BTBT	off-center spinning	BGTC	1 ppb	5.8 @ 1 ppb ($I_{ON}$)	2	−40	Yes	2017	[85]
Methanol	P3HT	spin coating	BGBC	570 ppm	3.5 @ 443 ppm ($I_{ON}$)	161	−1	No	2015	[70]
NH${}_{3}$	P3HT	spin coating	BGBC	1 ppm	25 @ 67 ppm ($I_{ON}$)	52	−1	Yes	2015	[70]
	P3HT:PS	spin coating	BGTC	5 ppm	52 @ 5 ppm ($I_{ON}$)	-	−40	Yes	2016	[98]
	P3HT	LB-nTM	BGTC	8 ppb	68.8 @ 1 ppm ($I_{ON}$)	-	−50	Yes	2017	[80]
	P3HT (NW)	spin coating	BGBC	20 ppm	24.2 @ 20 ppm ($I_{ON}$)	-	−45	Yes	2018	[81]
	P3HT (porous)	spin coating	BGTC	1 ppm	17.7 @ 10 ppm (μ)	-	−60	Yes	2019	[73]
	PQT-12:CdSe	FTM	BGTC	20 ppm	12.5 @ 20 ppm ($I_{ON}$)	65	−40	No	2018	[87]
	PBTTT	FTM	BGTC	330 ppb	12.52 @ 1 ppm (I${}_{ON}$)	26	−7	Yes	2017	[84]
	DNTT (porous)	evaporation	BGTC	10 ppb	73 @ 1 ppm ($I_{ON}$)	95	−10	Yes	2017	[83]
	Eu[Pc(SPh)${}_{8}$]${}_{2}$	quasi-LS	BGTC	15 ppm	14 @ 800 ppm ($I_{ON}$)	120	50	Yes	2018	[86]
	O(Si-Und-BTBT-Hex)${}_{2}$	LS	BGBC	50 ppb	37.5 @ 1 ppm ($I_{ON}$)	180	−40	Yes	2018	[72]
	PPI(-DMAENBA)-b-P3HT	spin coating	BGTC	10 ppb	28.6 @ 100 ppb ($I_{ON}$)	3–7	−80	No	2018	[82]
	PnTI	spin coating	BGTC	1 ppm	10–20 @50 ppm ($I_{ON}$)	60–180	−30	No	2018	[88]
	LGC-D148	spin coating	TGBC	1000 ppm	98.3 @ 1000 ppm ($I_{ON}$)	-	−80	No	2017	[99]
	DPP2T-TT (porous)	blade coating	BGTC	1 ppb	27.8 @ 1 ppb ($I_{ON}$)	1	−20	Yes	2017	[85]
	PDFDT	blade coating	BGTC	1 ppm	56.4 @ 10 ppm ($I_{ON}$)	180	−5	No	2017	[100]
NO${}_{2}$	P3HT:ZnO@GO	spin coating	BGTC	1 ppm	32 @ 1 ppm ($I_{ON}$)	300	−60	No	2017	[94]
	P3HT:PVK	spin coating	BGTC	139.3 ppb	687 @ 600 ppb ($I_{DS}$) §	300	−40	Yes	2018	[105]
	P3HT	spray coating	BGBC	10 ppm	320 @ 10 ppm ($I_{ON}$)	500	−50	Yes	2018	[106]
	CuPc	evaporation	BGTC	415 ppb	160,000 @ 5 ppm ($I_{ON}$)	600	−40	Yes	2017	[107]
	CuPc (NW)/Ph5T2	evaporation	BGTC	50 ppb	424 @ 10 ppm ($I_{ON}$)	1080	−15	Yes	2017	[92]
	Pentacene	evaporation	BGBC	1 ppm	22.7 @ 30 ppm ($I_{ON}$)	180	−40	No	2017	[103]
	TES-ADT	spin coating/SVA	BGTC	10 ppm	23.8 @ 30 ppm (I${}_{ON}$)	20	−10	No	2018	[97]
	TIPS-pentacene	spin coating	BGTC	200 ppb	539 @ 1 ppm ($I_{ON}$)	800	−40	Yes	2018	[101]
	TIPS-pentacene:PS	spin coating/SD	BGTC	1 ppm	8 @ 50 ppm ($I_{ON}$)	50	−10	No	2019	[95]
	TIPS-pentacene	off-center spinning	BGTC	1 ppm	44.3 @ 250 ppb ($I_{ON}$)	800	−40	Yes	2019	[102]
	TIPS-pentacene	spin coating	BGTC	1.93 ppb	1329 @ 1 ppm ($I_{DS}$) §	500	−4	Yes	2019	[96]
	PCDTBT †	spin coating	BGTC	1 ppm	14 @ 1 ppm ($I_{ON}$)	234	−50	Yes	2018	[104]
CO	PDPP4T-T-Pd(II)	FTM	BGTC	10 ppb	62 @ 1 ppm ($I_{ON}$)	10	−20	Yes	2019	[93]
H${}_{2}$S	CuPc (NW)	evaporation	BGTC	20 ppb	1088 @ 10 ppm ($I_{ON}$)	1800	−15	Yes	2017	[92]
	O(Si-Und-BTBT-Hex)${}_{2}$	LS	BGBC	10 ppb	60 @ 1 ppm ($I_{ON}$)	200	−40	Yes	2018	[72]
	PDPP4T-T-Hg(II)	FTM	BGTC	1 ppb	57 @ 1 ppm ($I_{ON}$)	10	−20	Yes	2019	[93]
	PSFDTBT	dip coating	BGTC	1 ppb	71–83 @ 1 ppm ($I_{ON}$)	5	−30	No	2016	[91]
H${}_{2}$O	P3HT	spin coating	BGBC	46 ppm	17 @ 249 ppm ($I_{ON}$)	298	−1	No	2015	[70]
	P3HT (porous)	spin coating	BGTC	100 ppm	35.7 @ 1000 ppm (μ)	-	−60	Yes	2019	[73]
	PBIBDF-BT	spin coating	BGTC	2858 ppm	99.8 @ 9146 ppm ($I_{ON}$)	0.44	−80	Yes	2017	[89]

When not an exact value, a gas sensing parameter is either the minimum (LoD and $t_{set}$) or maximum (*R* and *V*) value found in the cited publication. § Gas sensing parameters were extracted in the subthreshold regime or at $V_{GS}=0$ V for higher responsivity. † Poly[N-9${}^{\prime \prime }$-hepta-decanyl-2,7-carbazole-alt-5,5-(4${}^{\prime }$,7${}^{\prime }$-di-2-thienyl-2${}^{\prime }$,1${}^{\prime }$,3${}^{\prime }$-benzothiadiazole)]. ‡ BC = bottom contact; TC = top contact.

**Figure 9 materials-14-00003-f009:**
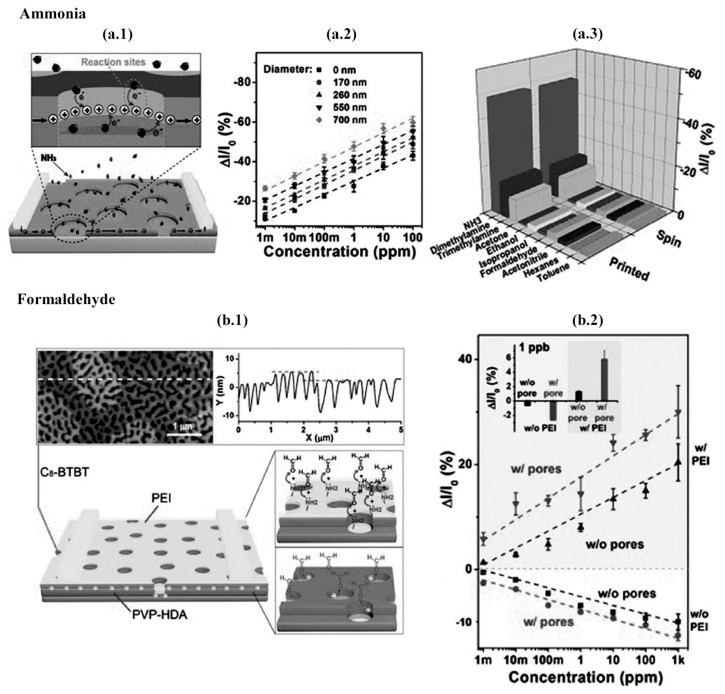
High-sensitivity gas sensors for ammonia and formaldehyde detection: (**a.1**) Schematic diagram of a porous DPP2T-TT TFT-based NH${}_{3}$ sensor. The magnified cartoon illustrates the charge transport reaction occurring at the conductive channel with NH${}_{3}$. (**a.2**) Current response to NH${}_{3}$ with concentrations ranging from 1 ppb to 100 ppm and pore sizes from 0 to 700 nm. (**a.3**) Sensor performance of transistors using printing and spin coating. All VOCs at 1 ppm. (**b.1**) Schematic diagram of a porous C8-BTBT TFT-based formaldehyde sensor. The atomic force microscopy figure and cross-sectional profile of the semiconductor film are shown. Formaldehyde interaction with the TFT is illustrated. (**b.2**) Current response of transistor with (inverted triangle) and without (triangle) pores with a PEI film as compared to pristine transistors with (circle) and without (square) pores to formaldehyde with concentrations ranging from 1 ppb to 1000 ppm. The inset shows the magnified current responses at 1 ppb. The pore size was ca. 500 nm. Reprinted from [85], with permission, from John Wiley and Sons, © 2017.

Formaldehyde (CH${}_{2}$O) is a common pollutant in the air and a carcinogen when exposed to concentrations of tens of ppb for long periods. Besides, formaldehyde is a biomarker for breast cancer, when concentrations above 1.2 ppm are exhaled in contrast to 0.3 ppm from a healthy person [144,166,167]. Zhang et al. achieved a selective gas sensing response to CH${}_{2}$O with a LoD of less than 1 ppb, 5.8% at 1 ppb, and a 2 s time response (see Figure 9b) [85]. In order to do that, the authors replaced DPP2T-TT with C${}_{8}$-BTBT by off-center spin coating and kept the same nanoporous structure previously used for NH${}_{3}$ detection. However, formaldehyde detection is challenging due to its low reactivity with most organic semiconductors. This limitation was circumvented by including a sensory layer of poly(ethyleneimine) (PEI), which is rich in primary amine groups. The PEI film donates electrons to the the hole-conducting C${}_{8}$-BTBT, which brings the transistor current down. When the carbonyl groups of CH${}_{2}$O reacts with PEI amine groups, the C8-BTBT layer is dedoped and the $I_{D}$ increases again.

Ethylene (C${}_{2}$H${}_{4}$) occurs naturally as a plant hormone to regulate physiologically important events (e.g., ripening and senescence) [168]. Besar et al. detected ethylene by blending P3HT with a porogenic compound, N-(tert-butoxy-carbonyloxy)-phthalimide, and Pd particles (< 1 μm diameter) [90]. The semiconducting layer was deposited by spin coating on oxidized Si substrates with pre-patterned Au electrodes. Whereas the porogenic material was responsible to increase the overall sensor surface area and expose π-electrons, Pd particles were used as receptors for ethylene molecules. Similar to H${}_{2}$ storage, ethylene binds strongly to transition metals forming stable complexes [169]. Although promising, further work is necessary to reach the sub-ppm level needed for monitoring ethylene during storage of fruits and vegetables.

According to Table 1, OTFT-based gas sensors for VOC detection feature a limit of detection from 1 ppb to 1100 ppm, a responsivity from 1.5 to 33.7%, a response time from 2 to 300 s and a p-type operating voltage from −1 to −60 V. In addition, they are usually poorly selective to one particular volatile organic compound. These detection limits are comparable to MOx chemosensors [170,171]. However, this performance is usually worse than those of photo-ionization detectors (PIDs), spectrophotometers, interferometers or fluorescent probes coupled to optical detectors (e.g., photodiode, CMOS or CCD), GC, GC/MS and proton transfer reaction quadrupole MS (PTR-QMS) [150,151,152,170,172,173].

#### 2.5.3. NO${}_{2}$

Another widely-investigated gas is nitrogen dioxide (NO${}_{2}$) [92,94,95,96,97,101,102,103,104,105,106,107]. This reddish- brown poisonous and environment pollutant gas with a pungent odor has many similarities to NH${}_{3}$, since it finds application in the manufacturing of fertilizers and explosives [174]. However, its electron-withdrawing character leads to the opposite effect on gas sensor response. According to Seo et al. [97], a positive current variation for a p-type OTFT upon exposure to NO${}_{2}$ is related to an increase in hole carrier density, and consequently, an increase in the effective mobility. In addition, grain boundaries from the solvent vapor annealing in 1,2-dichloroethane provided a pathway for the diffusion of the gaseous analyte. Polycrystalline TES-ADT TFTs featured a 23.8% response at 30 ppm of NO${}_{2}$ with a 20 s response time. Although selectivity was not discussed, these devices went through a UV-ozone (UVO) treatment of SiO${}_{2}$ prior to thin-film deposition. Huang et al. demonstrated 160,000% NO${}_{2}$ response at 5 ppm of CuPc TFTs with the dielectric surface submitted to UVO treatment, which was ca. 400 times greater than for those without treatment [107]. Pristine organic dielectrics have a low polarity surface with carbon–hydrogen bonds, and therefore, negligible binding affinity for NO${}_{2}$ molecules. After the UVO treatment, oxygenated functionalities are introduced on the dielectric’s surface, thereby enhancing the adsorption of polar molecules via either hydrogen bonding or van der Waals interactions [175]. In addition, water molecules in the atmosphere react with NO${}_{2}$ + O${}_{2}$ to form nitric acid. Nitric acid can act as a dopant for the semiconductor film at the interface with the gate dielectric. Despite the high sensitivity, both onset and recovery times were not fast (>600 s). Authors suggested increasing the working temperature to quicken the response. The devices were selective for NO${}_{2}$ in the presence of common gas pollutants, such as SO${}_{2}$, NH${}_{3}$, H${}_{2}$S and CO${}_{2}$. In a recent work, Shao et al. achieved a LoD of 1.93 ppb for an OTFT as NO${}_{2}$ sensor [96] (see Figure 10a). The device was based on TIPS-pentacene semiconductor and PMMA dielectric, both deposited by spin coating on glass/ITO and Si/SiO${}_{2}$ substrates with top Au electrodes evaporated through a shadow mask.

**Figure 10 materials-14-00003-f010:**
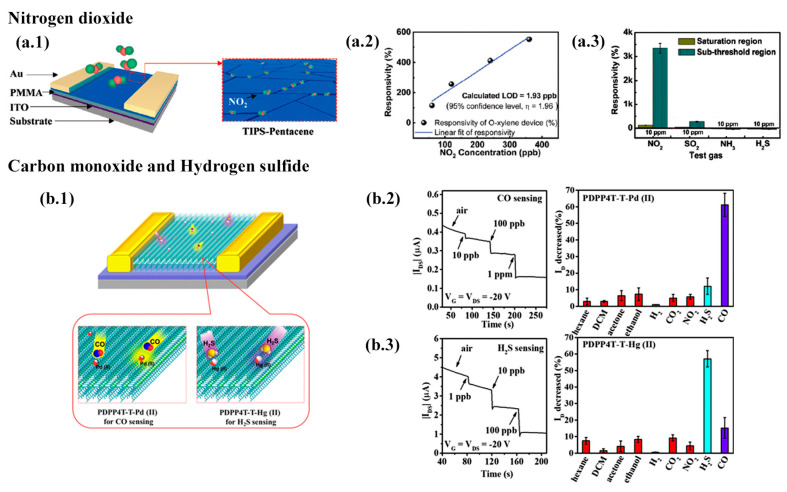
High-sensitivity gas sensors for nitrogen dioxide, carbon monoxide and hydrogen sulfide detection: (**a.1**) Schematic diagram of the TIPS-pentacene TFT-based NO${}_{2}$ sensor from an o-xylene solution. Illustration of the gas sensing mechanism. (**a.2**) Plot for the limit of detection (LoD) calculation. (**a.3**) Sensor response in the saturation and subthreshold regions towards 10 ppm of NO${}_{2}$, SO${}_{2}$, NH${}_{3}$ and H${}_{2}$S. Republished from reference [96] with permission from The Royal Society of Chemistry. (**b.1**) Schematic diagrams of PDPP4T-T-based TFTs for CO and H${}_{2}$S sensing. (**b.2**) PDPP4T-T-Pd(II)-based TFTs. Left graph: Current response to CO with concentrations ranging from 10 ppb to 1 ppm. Right graph: Sensor performance towards hexane (52,000 ppm), dichloromethane (DCM) (301,000 ppm), acetone (1000 ppm), ethanol (1200 ppm), H${}_{2}$ (pure), CO${}_{2}$ (pure), NO${}_{2}$ (100 ppm), H${}_{2}$S (100 ppm) and CO (1 ppm). (**b.3**) PDPP4T-T-Hg(II)-based TFTs. Left graph: Current response to H${}_{2}$S with concentrations ranging from 1 to 100 ppb. Right graph: Sensor performance towards hexane (52,000 ppm), DCM (301,000 ppm), acetone (1000 ppm), ethanol (1200 ppm), H${}_{2}$ (pure), CO${}_{2}$ (pure), NO${}_{2}$ (100 ppm), H${}_{2}$S (1 ppm) and CO (100 ppm). Reprinted with permission from reference [93]. Copyright (2019) American Chemical Society.

The low LoD is partially due to a careful tuning of semiconducting thin-film morphology (i.e., uniform film crystallinity and adequate grain boundary dimensions) by proper solvent choice. For example, sensor response to NO${}_{2}$ was enhanced to about 58 times when the semiconductor material was processed in o-xylene instead of chlorobenzene. In addition, authors compared the performance in two different operation regimes: subthreshold and saturation. TIPS-pentacene from o-xylene solutions showed a responsivity of 1329% at 1 ppm NO${}_{2}$ in the subthreshold region, which was more than 18 times greater than the performance achieved in the saturation region (ca. 71%). According to authors, operation in the subthreshold regime should make the detection of ultra-low NO${}_{2}$ concentrations possible. In that study, the OTFTs showed selectivity toward NO${}_{2}$ when compared to other pollutant gases often found in exhaust gases. Further improvements might involve adding interface metal oxides as molybdenum oxide (MoO_x_) to facilitate charge carrier injection from gold contacts [103].

In general, OTFTs’ response to NO${}_{2}$ in Table 1 show a limit of detection from 1.93 ppb to 10 ppm, a responsivity from 8 to 160,000%, a response time from 20 to 1080 s and an operating voltage from −4 to −60 V using p-type semiconductors. Most studied OTFTs are selective to NO${}_{2}$. The detection limits are higher than that achieved with colorimetric sensors, but comparable to quartz-enhanced PAS (QEPAS) and H-type longitudinal resonant photoacoustic cells [147,176,177]. It is, however, less performant than Faraday rotation spectroscopy (FRS), a costly and sophisticated technique which uses the magnetic circular birefringence (MCB) effect [177].

#### 2.5.4. H${}_{2}$S and CO

Hydrogen sulfide (H${}_{2}$S) is a colorless toxic and flammable gas with an odor of rotten eggs. Long-term exposure to concentrations as low as 5 ppb can lead to respiratory, eye and nasal diseases. In order to detect even lower concentration levels, Lv et al. fabricated a poly[2,7-(3${}^{\prime }$,6${}^{\prime }$-dioctyloxy)-9,9${}^{\prime }$- spirobifluorene-alt-5,5-(4${}^{\prime }$,7${}^{\prime }$-di-2-thienyl-5${}^{\prime }$,6${}^{\prime }$-dioctyloxy-2${}^{\prime }$,1${}^{\prime }$,3${}^{\prime }$- benzothiadiazole)] (PSFDTBT) TFT by dip-coating onto OTS-modified Si/SiO${}_{2}$ wafers with top evaporated Au electrodes. By optimization of the active layer thickness, the best gas sensors featured a 1 ppb LoD, 71–83% response at 1 ppm and 5 s response time [91]. It was verified that current variation was the highest for 20 nm-thick films, whereas 5 and 25 nm-thick films provided the worst performance. They hypothesized that the rates of H${}_{2}$S absorption/desorption change differently for each semiconductor film thickness. A thick film is not preferred, since the semiconductor/dielectric interface is buried under that thickness. As the thickness decreases, desorption of H${}_{2}$S molecules is faster than absorption. Therefore, a too thin active layer film may not help to retain enough analyte molecules during the measurement. PSFDTBT films showed p-type behavior upon exposure to H${}_{2}$S. There was a current decrease, which was ascribed to hole trapping from electron pairs of H${}_{2}$S molecules.

Carbon monoxide (CO) is a colorless poisonous gas produced from burning fuels. It is, however, odorless, unlike NH${}_{3}$, NO${}_{2}$ and H${}_{2}$S, making it especially dangerous. Usually found in ppm levels as an air pollutant, safe levels are in the hundreds of ppb [178]. In a recent work, Yang et al. were capable of detecting 10 ppb and 1 ppb of CO and H${}_{2}$S, respectively, by coordinating thymine groups in the side chains of PDPP4T-T with metal ions (see Figure 10b) [93]. The floating film transfer method was used to obtain the composite thin-film on pre-patterned Au electrodes and OTS-treated Si/SiO${}_{2}$ substrates. Metal ions were incorporated into the film by dropping 50 μL of PDPP4T-T in chloroform onto an aqueous solution surface containing either K${}_{2}$PdCl${}_{4}$ or Hg(ClO${}_{4}$)${}_{2}$. Pd and Hg ions acted as reactive sites for sensing CO and H${}_{2}$S, respectively. H-bonding among the thymine groups were also responsible to strengthen the interchain interactions with μ in crystalline PDPP4T-T reaching 9.1 cm${}^{2}$/V·s. Despite the high selectivity towards these gases, since the sensing reactions were irreversible, the gas sensors were considered not reusable.

Results displayed in Table 1 regarding H${}_{2}$S and CO show a limit of detection ranging from 1 to 20 ppb, responsivity of 57 to 1088% and response time of 5 to 1800 s. These devices are p-type OTFTs operating at the maximum voltage of −15 to −40 V and exhibiting high selectivity towards H${}_{2}$S and CO. OTFTs outperform MOx chemosensors and show comparable performances to SAW sensors, colorimetric sensors and fluorescent probes coupled to optical detectors, QEPAS, micro-electro-mechanical systems (MEMS)-based sensors and PTR-QMS [147,148,149,150,152,179].

## 3. Outlook

### 3.1. Future Prospects

Future work in the field of organic thin-film transistors as gas sensors will keep integrating semiconducting films from multiple or single crystals with p-type behavior and charge carrier mobilities ranging from $10^{-3}$ to 10 cm${}^{2}$/V·s. Ambipolar and stable compounds (e.g., copolymers with donor and acceptor moieties) based on widely studied p-type organic semiconductors still need to have their behavior in a gas sensing device investigated. Well-known and consolidated fabrication techniques such as spin-coating and thermal evaporation will be gradually replaced by more innovative processes, such as meniscus-guided coating and transfer methods. As previously described, these new processes waste ten times less material and lead to larger crystal dimensions (> 100 μm). Among the possible OTFT structures, bottom gate structures will remain as the most suitable, since this configuration exposes the transistor channel at the semiconductor/dielectric interface. Top contacts are preferred for decreasing the contact resistance and producing larger semiconducting crystals. Patterning techniques on top of organic films need to be incorporated into the fabrication methodology to take advantage of such structures. Potentially low-cost, large-area, flexible devices might integrate organic conductors to replace widely-used vacuum evaporation of metals. Although the performance of a transistor is easily monitored through an on-to-off voltage swing, this task can be complicated for a portable device. Practical implementations demand gas sensors operating on battery power, and therefore, low voltages. Drain-to-source current is already the most monitored device parameter. Evaluating the current variation in the subthreshold regime will be a feasible strategy to enhance sensitivity. As mobility and threshold voltage calculations require wide current versus voltage scans, demanding far more computing capabilities, these parameters will be mostly used as accessory tools to understand the sensing mechanism of a device. A multiparametric analysis is typical in electronic noses. This kind of system is aimed at the detection of volatile organic compounds. Individual organic transistors shall focus more on air-pollutants or toxic gases (e.g., NH${}_{3}$, NO${}_{2}$, CO and H${}_{2}$S). The incorporation of porogenic materials and composites with metal or inorganic semiconductors will be among the main strategies to improve performance and achieve even lower limits of detection. Finally, stability still remains as one of the major issues to be addressed for highly-sensitive reusable devices in the near future. In summary, organic thin-film transistors have a promising future in gas sensor applications.

### 3.2. Conclusions

This publication reviewed a small number of organic thin-film transistor technologies, trying to bring together some of the current strategies put forward to improve gas sensing performance since 2015, alongside widespread fabrication and performance evaluation techniques. Besides its detailed illustrations and careful explanations, this manuscript can also be used as an index due to an extensive list of references. 

## Figures and Tables

**Figure 2 materials-14-00003-f002:**
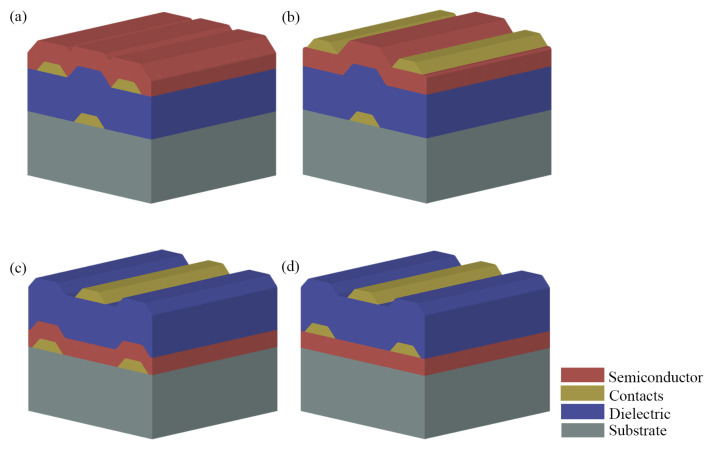
Organic thin-film structures: bottom gate, (inverted) (**a**) bottom contact (coplanar) and (**b**) top contact (staggered); top gate, (**c**) bottom contact and (**d**) top contact. Note that the device is not fully scaled, since the substrate thickness can vary from less than a micron to more than a millimeter. Stacked films are not necessarily flat and conformability depends on the deposition techniques applied.

**Figure 3 materials-14-00003-f003:**
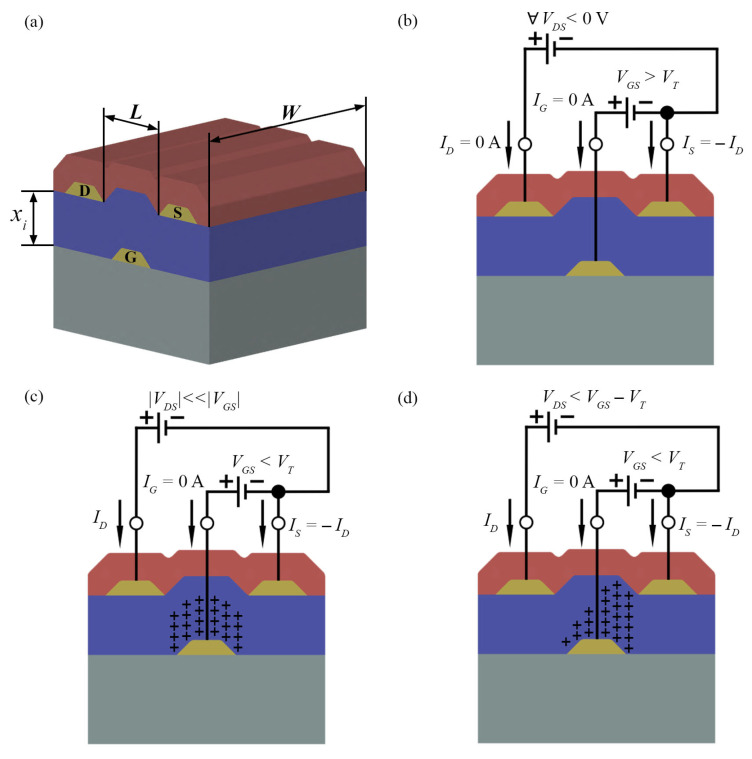
P-type field-effect-transistor (FET): (**a**) structural parameters and device electrodes; (**b**) cut-off (**c**) triode and (**d**) saturation operating modes.

**Figure 5 materials-14-00003-f005:**
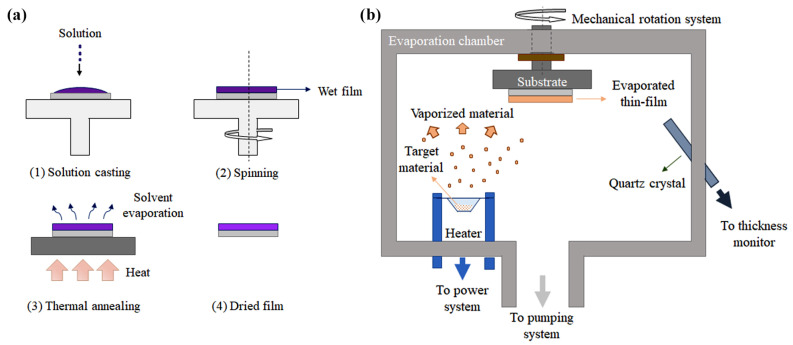
Illustration of well-established and pioneering techniques for thin-film deposition of organic electronic devices: (**a**) spin coating and (**b**) thermal evaporation.

**Figure 7 materials-14-00003-f007:**
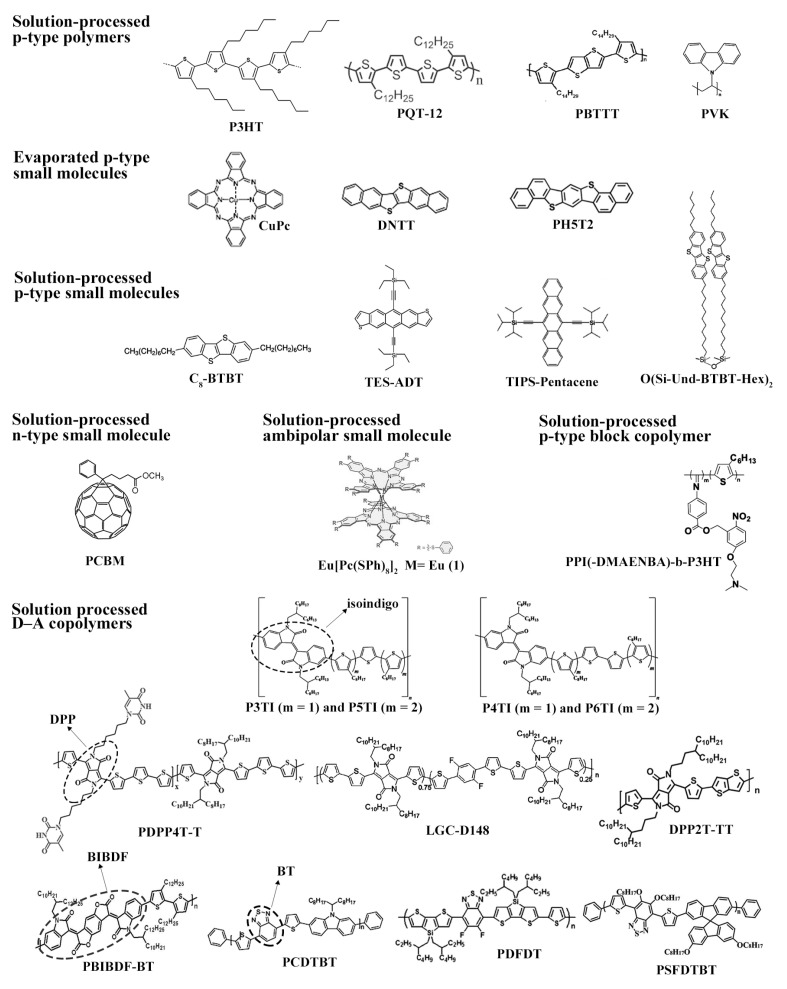
Chemical structures of organic semiconducting molecules for organic thin-film transistors in gas sensing applications.

**Figure 8 materials-14-00003-f008:**
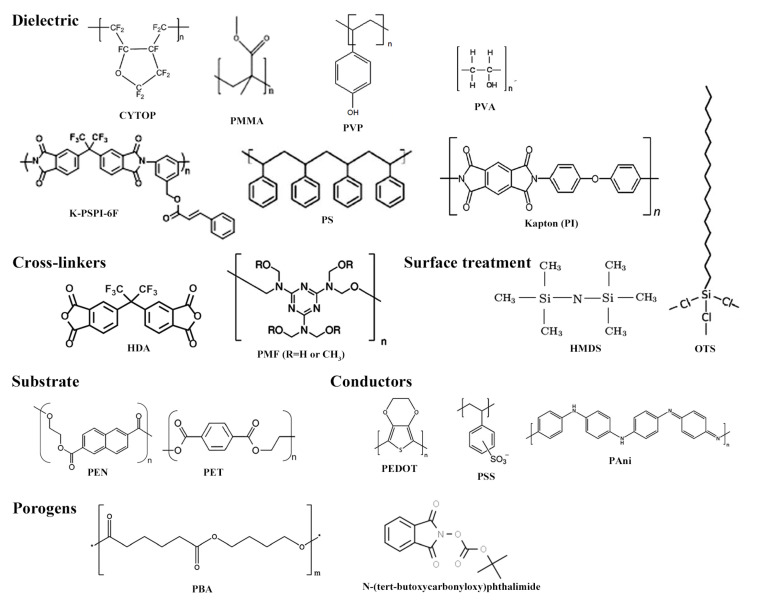
Chemical structures of organic molecules used as dielectrics, conductors, surface treatments and substrates for organic field effect transistors in gas sensing applications.

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
