# Peer review of "Organic Thin-Film Transistors as Gas Sensors: A Review"

_materials, 2020, doi:10.3390/ma14010003_

Round 1
Reviewer 1 Report
This paper reviews recent development (since 2015) in the studies of organic thin-film transistors for gas-sensing applications. I think that the paper should be useful for many researchers who work on or are interested in the topic. The paper is well-organized and written, and I think that the paper is suitable for the publication in ”Materials”. Before the publication, I would like to ask the authors to take following corrections.
Line 48: The following description is not correct in terms of the use of ”polymers”.
"The following decades have witnessed the increase in charge carrier mobility in OTFTs as a result of new processing methods and the synthesis of myriad new polymers."
The authors can use ”small-molecule semiconductors, polymeric semiconductors, and nanocarbons”, for instance.
line 55: I think that the use of ”allotropic forms of carbon” is not appropriate in this context.
line 60: The following sentences are simply illogical and confusing. This is because of the unrelated description of graphene for the review.
”A new breakthrough was achieved with graphene in 2004, when a strong ambipolar electric field effect was observed, reaching mobilities of approx. 10,000 cm2 /Vs [7]. After years of continuous improvements, the best OTFTs reported today [25] reach a charge carrier mobility of ca. 1-10 cm2 /Vs, which is, however, about 100 to 1,000 times lower than that achieved with single-crystal silicon. ”
line 104: The following sentences are confusing. This is because the authors do not use absolute value for the comparison between VGS and VT.
”The first operating mode, i.e. cut-off shown in Figure 3(b), is defined for VGS > VT, in which VGS is not negative enough to form a conducting path between source and drain, the channel is depleted of holes and the drain-to-source current (ID) is zero.”
line 418: In related to the following descriptions, the authors should describe the comparison of the limit of detection (LoD) in the other gas-sensing techniques.
”It is worth noticing that there are several methods available for gas detection exhibiting selectivity and sensitivity, including gas chromatography, mass spectrometry, optical chemical sensing, and mass sensing [135-138].”
Figures: The authors should redraw Figures 2 and 3. It is not easy to understand the difference between the devices because of the dark and considerably deformed components. The used aspect ratios and rugged film surfaces are also misleading for the readers.
Author Response
Please, see the attachment.

Reviewer 2 Report
See attached.

Author Response
Please, see the attachment.

Reviewer 3 Report
Manuscript Number : materials-1002485
I recommend acceptance of the manuscript after minor revision with following comments:
The manuscript entitled ‘ORGANIC THIN-FILM TRANSISTORS AS GAS SENSORS: A REVIEW’ is representing a systematic review study on the application of organic thin film transistor (OTFT) as gas sensor. The first section of the manuscript contains a comprehensive review on the working principle of OTFTs for gas sensing, a concise description of devices’ architectures and parameter extraction based upon a constant charge carrier mobility model. Then, it moves on with methods of device fabrication and physicochemical description of main organic semiconductors recently applied to gas sensors (i.e. since 2015 but emphasizing even more recent results). Finally, it describes the achievements of OTFTs on detection of important gas pollutants alongside an outlook to the future of this exciting technology. It is well written and highly useful to examine the different aspects that require further attention of OTFTs for gas sensing applications. The manuscript fits the journal´s scope, it is of interest.
Here are some of points for further improvement:
- Some more lines can be added in the last paragraph of the introduction section (i.e. page number 2, line number 72) to describe the importance of the development of gas sensors in near future.
- Author should add one paragraph at the end of section 2.5.1, 2.5.2, 2.5.3, and 2.5.4, to compare the results of table 1.
- The outlook section can be divided into two sections such as future prospects and conclusions.
- The font size of the characters of Figure 9. (a.2), (a.3), (b.2) and Figure 10. (a.2), (a.3), (b.2) (b.3) should be enlarged.
Author Response
Please, see the attachment.
